nanotechnology

osteosarcoma, IGFBP$_5$, biosafety, superparamagnetic chitosan iron oxide nanoparticles, magnetic transfection

**Authors for correspondence:**
Xiangyang Qu
e-mail: drqxy04090@126.com
Dianming Jiang
e-mail: 201296@hospital.cqmu.edu.cn

# Magnetic transfection with superparamagnetic chitosan-loaded IGFBP$_5$ nanoparticles and their *in vitro* biosafety

Yue Tang[1,2,3], Jun Wu[2], Yuan Zhang[2], Lingpeng Ju[2], Xiangyang Qu[2] and Dianming Jiang[1,3]

[1]Department of Traumatic Joint Center, The Third Affiliated Hospital of Chongqing Medical University (General Hospital), No 1 Shuanghu Road, Yubei District, Chongqing 401120, People's Republic of China
[2]Department of Orthopedics, Children's Hospital of Chongqing Medical University, National Clinical Research Center for Child Health and Disorders, Ministry of Education Key Laboratory of Child Development and Disorders, Chongqing Key Laboratory of Pediatrics, Laboratory of Biomaterials, 136# Zhongshan 2 road, Yuzhong District, Chongqing 400014, People's Republic of China
[3]Department of Orthopedics, The First Affiliated Hospital of Chongqing Medical University, No 1 Medicine Road, Yuzhong District, Chongqing 400016, People's Republic of China

YT, 0000-0003-1470-3877; DJ, 0000-0001-9935-7066

We prepared the superparamagnetic chitosan nanoparticles (SPCIONPs) to study the application of them as gene vectors using a magnetic transfection system for the targeted treatment of lung metastasis of osteosarcoma. The SPCIONPs were characterized by transmission electron microscopy, Fourier transform infrared spectrometry, superconducting quantum interference device and atomic force microscopy. Their biosafety was determined by cell counting kit-8 (CCK8) and live–dead staining assays. The transfection *in vitro* was detected by laser confocal microscopy. SPCIONPs, which can bind closely to plasmids and protect them from DNA enzyme degradation, were prepared with an average particle size of approximately 22 nm and zeta potential of 11.3 mV. The results of the CCK8 and live–dead staining assays showed that superparamagnetic chitosan nanoparticles loaded with insulin-like growth factor-binding protein 5 (SPCIONPs/pIGFBP$_5$) induced no significant cytotoxicity compared to the control group. The result of transfection *in vitro* suggested that pIGFBP$_5$ emitted a greater amount of red fluorescence in the SPCIONPs/pIGFBP$_5$ group than that in the chitosan-loaded IGFBP$_5$ (CS/pIGFBP$_5$) group. In conclusion, the prepared SPCIONPs had good biosafety and could be effectively used to transfer pIGFBP$_5$ into 143B cells, and they thus have good application prospects for the treatment of lung metastasis of osteosarcoma.

This article has been edited by the Royal Society of Chemistry, including the commissioning, peer review process and editorial aspects up to the point of acceptance.

# 1. Introduction

Osteosarcoma is the most common primary bone malignancy in children and young people, with lung metastases occurring at the time of diagnosis in approximately 15–20% of patients [1]. At present, a multimodal treatment approach has been adopted for osteosarcoma treatment, with a methotrexate, doxorubicin and cisplatin (MAP) regimen as the first choice. The treatment is mainly based on chemotherapy combined with limb salvage or amputation, but the clinical therapeutic effect is still unsatisfactory. The 5-year survival rate for patients with osteosarcoma is approximately 70%, while the survival rate for patients with metastatic or recurrent disease is extremely low, with an overall survival rate of less than 20% [2]. Therefore, gene therapy has become a trend in osteosarcoma treatment [3].

Insulin-like growth factor-binding protein 5 (IGFBP$_5$) is a member of the insulin-like growth factor (IGF) system, which plays an important role in cell growth, differentiation and apoptosis [4]. IGFBP$_5$ primarily binds to insulin growth factors to inhibit the IGF1 and IGF2 signalling pathways, thereby inhibiting the proliferation, migration and invasion of osteosarcoma, and is a key inhibitor of *in situ* osteosarcoma growth and lung metastasis [5]. Our group previously used adenovirus to induce IGFBP$_5$ overexpression in osteosarcoma cells and confirmed that IGFBP$_5$ can induce apoptosis and inhibit invasion, migration and proliferation of osteosarcoma cells [6].

Non-viral gene vectors, which are currently a focus of research, will not cause immune responses, potential immunogenicity and lethal inflammation, in contrast to adenovirus vectors [7,8]. Non-viral vectors commonly include liposomes, polyethyleneimine (PEI) and chitosan (CS). Cationic liposomes, which have higher transfection rates but are more toxic and expensive, are the most commonly used liposomes. Cationic liposomes also have no sustained-release effect and thus require repeated administration during experiments *in vivo*, and they are greatly affected by serum [9]. PEI is currently the most commonly used and most effective non-viral vector due to its strong proton-buffering capacity. PEI is the gold standard for polymeric gene delivery vectors. The proton sponge effect can cause endosomes to swell and release the plasmid into the cytoplasm, which allows the plasmid to enter the nucleus and complete gene expression [10,11]. Therefore, PEI was used as a positive control group in this study. However, due to its severe toxicity, including mitochondrial damage and apoptosis, the *in vivo* applications of PEI are limited [12,13]. CS is currently used in tissue engineering, drug and gene delivery, wound healing, and antibacterial and anti-tumour applications because of its physiological and biological activity; biocompatibility with a variety of organs, tissues and cells; low toxicity, with generally recognized as safe (GRAS) approval by the Food and Drug Administration (FDA) [14–16] and compatibility with chemical or enzymatic modifications. In addition, CS can be dissolved in acetic acid to condense the plasmid and prevent degradation by DNase, and it is a good substitute for viral vectors, but unmodified CS has a low transfection rate as a gene vector [17].

Superparamagnetic iron oxide nanoparticles (SPIONPs) can act as a reagent with both active and passive targeting effects and can concentrate the carried drugs in the target tissue under the effect of an external magnetic field. Because of their advantages, such as biocompatibility, stability, environmental safety and low price, SPIONPs are widely used in biomedicine, including for targeted drug delivery [18–21], bioimaging, thermotherapy, photoablation therapy, biosensing and thermal olfaction [22–25]. At present, there are many studies to modify the surface of SPIONPs so that its surface is distributed in different groups, such as polymers, biomolecules, silica and metals which can provide the ensemble functional reactive group, e.g. aldehyde groups, hydroxyl groups, carboxyl groups and amino groups. Their groups can be linked to antibodies, proteins, DNA, enzymes and other bioactive substances for further application [26]. Therefore, in this study, we prepared SPIONPs coated with CS to generate superparamagnetic chitosan-coated iron oxide nanoparticles (SPCIONPs) and conducted experimental studies to investigate their *in vitro* biosafety and use for cell transfection in the presence of magnetic transfection system.

# 2. Material and methods

## 2.1. Materials

Citric acid, ferric chloride hexahydrate, ferrous sulfate heptahydrate, 25–28 wt% ammonia solution and 1-(3-dimethylaminopropyl)-3-ethylcarbodiimide hydrochloride (EDC) were purchased from Macleans (Shanghai, China); Dulbecco's Modified Eagle Medium (DMEM) and trypsin were purchased from Gibco (CA, USA); chitosan (50 kDa, 98% polyacetyl), *n*-hydroxysuccinimide (NHS) and fluorescein

isothiocyanate (FITC) were purchased from Sigma-Aldrich (MO, USA); fetal bovine serum was purchased from PAN-Biotech (Adenbach, Germany); the Prussian blue staining kit was purchased from Solarbio (Shanghai, China); the DNA marker was purchased from Tiangen (Beijing, China); Hoechst, DNase 1 and LB broth were purchased from Beyotime Biotech (Beijing, China); and the Endo-free Plasmid Maxi Kit-25 was purchased from OMEGA (GA, USA).

## 2.2. Preparation of SPCIONPs

Ferric chloride hexahydrate (2 g) and ferrous sulfate heptahydrate (1.4 g) were dissolved in 5 ml of water, followed by ultrasonication until the chemicals were completely dissolved. The solution was then transferred to a three-necked flask and stirred under nitrogen. Subsequently, 20 ml of 0.1% citric acid solution was added to the solution, which was then stirred for 10 min. The three-necked flask was then placed in a water bath at 85°C. A total of 20 ml ammonium hydroxide was added with a dropping funnel; the addition was completed within 10 min, and stirring was continued for another 30 min [27,28]. The resulting suspension was centrifuged at 8500 r.p.m. for 10 min, the sediment was discarded, and the supernatant was removed to obtain the ferrofluid of citric acid-coated magnetite particles (SPIONPs). Finally, 5 ml of the ferrofluid of citric acid-coated magnetite particles was added to MES buffer solution with a pH of 5, and EDC and NHS were subsequently added under stirring to activate the carboxyl group on the surface of the ferrofluid of citric acid-coated magnetite particles, followed by the addition of 5 ml of 20 mg ml$^{-1}$ 50 kDa CS solution in acetic acid and stirring at 300 r.p.m. at room temperature for 6 h [29]. The liquid was collected after 48 h of dialysis with a dialysis bag (retained molecular weight over 50 000). A portion of the liquid was lyophilized and stored at 4°C for later use.

## 2.3. Characterization

The size and morphology of the prepared SPCIONPs were observed with a transmission electron microscope (TEM, HITACHI, USA). The size distribution was determined by measuring diameters of 100 NPs randomly selected on the TEM micrographs. A Zetasizer Nano ZS (Malvern, UK) was used to determine the zeta potential of the CS, SPIONPs, SPCIONPs and SPCIONPs/pIGFBP$_5$. The magnetic induction of SPIONPs and SPCIONPs was measured at 300 K using a superconducting quantum interference device (SQUID, a magnetic property measurement system (MPMS), Quantum Design, USA). The structural characterization of CS, SPIONPs and SPCIONPs was performed with a Fourier transform infrared (FT-IR) spectrometer using the KBr compression method. The molecular structure of prepared SPCIONPs were loaded onto the XY scanner of the AFM (IPC-208B, Chongqing University, China) [30–32]; the area to be scanned was localized on the monitor, and then images generated at room temperature under ambient conditions using the non-contact mode. A 100 mm scanner and a STM probe were used in the study, with a customized tungsten filament as the microcantilever to detect an area of 12.01 × 12.01 nm.

## 2.4. Agarose gel electrophoresis

Agarose gel electrophoresis was used to detect the ability of SPCIONPs to condense the plasmids. The SPCIONPs and plasmids (1 µg) were mixed based on N/P ratios of 10/1, 5/1, 2.5/1, 2/1, 1/1, 1/2, 1/2.5, 1/5 and 1/10. A 0.8% agarose gel was prepared with 1% Tris-acetate-EDTA (TAE) buffer, and the samples were electrophoresed at 85 mV for 30 min. The optimal N/P ratios of SPCIONPs and pIGFBP$_5$ were observed. The naked plasmid was used as the control.

## 2.5. Protection assay

After incubation of SPCIONPs with the plasmids at different N/P ratios, 1 µl of DNase I (corresponding to 1 µg of plasmid) was added to 10× reaction buffer [100 mM Tris–HCl (pH = 7.5 at 25 mM), 25 mM MgCl$_2$, 1 mM CaCl$_2$)]. After 15 min of incubation in a 37°C water bath, 1 µl of ethylenediaminetetraacetic acid (EDTA) (25 mM, pH = 8) was added, followed by incubation at 65°C for 15 min to inactivate DNase I. The naked plasmid was used as the control. A 0.8% agarose gel was prepared with 1% TAE, and the samples were electrophoresed at 85 mV for 30 min.

## 2.6. Prussian blue staining

Prussian blue can stain iron atoms in cells. First, 143B cells were seeded in 12-well plates at $1 \times 10^5$ cells/well. After the cells became adherent, SPCIONPs/pIGFBP$_5$ were added, and a magnetic field was applied for 30 min. After 4 h, the cells were washed three times with phosphate-buffered saline (PBS) and fixed in paraformaldehyde for 20 min. Perls solution was prepared at a 1 : 1 ratio, and 200 µl of the solution was added to each well, followed by incubation at 37°C for 40 min. Cells were washed with ddH$_2$O, and 200 µl of neutral red solution was added and incubated with the cells at room temperature for 2 min. After the staining solution was rinsed with ddH$_2$O, the cells were observed under a light microscope.

## 2.7. *In vitro* toxicity assay

The CCK8 reagent was used to calculate the cell survival rate in order to detect cytotoxicity. First, 143B cells were seeded in 96-well plates at $4 \times 10^3$ cells/well with five duplicated wells in each group. After the cells became adherent, the reagents were added to each of the following groups: the control group, the PUC19 group (the PUC19 plasmid was used to clone *Escherichia coli* without the target gene), the CS/PUC19 group, the SPCIONPs/PUC19 group and the PEI/PUC19 group, and a magnetic field was applied for 30 min. A total of 10 µl CCK8 reagent was added at 24 and 48 h, followed by incubation at 37°C for 3 h. The optical density (OD) at 450 nm was detected using a microplate reader, and the cell survival rate was calculated as follows: cell survival rate = (experimental group − blank group)/ (control group − blank group) ∗ 100%.

## 2.8. Live–dead staining

The 143B cells were seeded in a 24-well plate at $5 \times 10^4$ cells/well. After the cells became adherent, reagents were added to each of the following groups: the control group, the PUC19 group, the CS/PUC19 group, the SPCIONPs/PUC19 group and the PEI/PUC19 group. The magnetic field was applied for 30 min. The cells were collected by centrifugation and resuspended in a mixture of 1 µl A (1 mM Live-Dye) + 1 µl B (1 mg ml$^{-1}$ PI) + 1 ml staining buffer at 24 h and 48 h. After incubation at 37°C for 15 min in the dark, the cells were observed under a fluorescence microscope. The following method was used to calculate the cell survival rate: number of red cells/(number of red cells + number of green cells) ∗ 100%.

## 2.9. Cell uptake

The 143B cells were seeded in a 34 mm confocal dish at $1 \times 10^5$ cells/well, and FITC-CS/pIGFBP$_5$ and FITC-SPCIONPs/pIGFBP$_5$ were added after the cells became adherent. pIGFBP$_5$ is an expression plasmid with red fluorescence protein (RFP). After a magnetic field was applied for 30 min, the non-endocytosed particles were washed away with PBS, and cell culture was continued for 6 or 24 h. At each time point, the cells were washed with PBS and stained with Hoechst for 20 min. After washing with PBS, the medium was added, and the cells were observed under a confocal microscope.

## 2.10. *In vitro* transfection assay

The 143B cells were seeded in a 12-well plate, and when the confluency of cells reached 80–90%, the medium was aspirated, and the cells were washed three times with PBS. A total of 800 µl of supplement-free medium was added to each well, and 4 nM HCl was used to prepare serum-free medium at a pH of 5.5. The CS/pIGFBP5 and SPCIONPs/pIGFBP5 was used serum-free medium to dilute to 200 µl per well, and the mixtures were incubated at 50°C for 20 min and used to transfect cells; 200 µl PEI was added to the PEI group (positive control group), and a magnetic field was applied for 30 min. After 4 h of transfection, the cells were washed three times with PBS, cultured for 24 h, and observed under an inverted fluorescence microscope.

## 2.11. Statistical analysis

All analyses were performed using IBM SPSS 20.0 (software version 20.0, SPSS Inc., Chicago, IL, USA) or GraphPad Prism (version 7.00, GraphPad Software, Inc., San Diego, CA, USA). Multigroup comparisons were performed using ANOVA. A value of $p < 0.05$ was considered statistically significant.

# 3. Results

## 3.1. Characterization

SPCIONPs were successfully prepared by the dehydration condensation method, and their physico-chemical characterizations were shown in figure 1. The morphology of SPCIONPs was observed by TEM, and the average particle size of a single particle was about 22 nm (figure 1a,b). Figure 1c shows that the zeta potentials of CS, SPIONPs and SPCIONPs were 42.9, −27.9 and 27.1 mV, respectively. The negatively charged plasmids were combined through electrostatic adsorption to form SPCIONPs/pIGFBP$_5$, which were positively charged with a zeta potential of 11.3 mV.

A comparison of transverse magnetic induction results (figure 1d) indicated that the superparamagnetic behaviour of SPCIONPs was weaker than that of SPIONPs. Both curves intersected the origin, and the hysteresis was zero. Therefore, both were superparamagnetic, which was consistent with previous research results [29,33].

The infrared spectra of CS, SPIONPs and SPCIONPs in the wavenumber range of 400–4000 cm$^{-1}$ are shown in figure 1e. The CS spectrum had unique amino (NH$_2$) characteristic peaks at 3343.63 and 1617.23 cm$^{-1}$. The spectra of SPIONPs had characteristic COOH peaks, with C=O at 1646.05 cm$^{-1}$ and O–H at 3426.82 cm$^{-1}$, the characteristic peaks of Fe$_3$O$_4$ and Fe–O were observed at 569.91 cm$^{-1}$. SPCIONPs exhibit the amide bond (NH–CO) formed by the amino group of chitosan and the carboxyl group of SPIONPs, and the characteristic peaks of N–H (3439.63 cm$^{-1}$ and 1617.23 cm$^{-1}$), C=O (1687.69 cm$^{-1}$) and Fe–O (579.36 cm$^{-1}$) were not affected (figure 1e).

AFM also verified that CS was bound to SPIONPs by amide bonds. White dots formed a six-membered ring representing the basic structure of CS; black dots represented Fe atoms; and red and white dots bound together represented N–H and C=O (figure 1f).

## 3.2. Plasmid binding assay

After the addition of samples at N/P ratios of 10/1, 5/1, 2.5/1, 2/1, 1/1, 1/2, 1/2.5, 1/5 and 1/10, we found that when the N/P was 2/1, the plasmids did not run out of the electrophoresis lanes, which confirmed that SPCIONPs could fully bind to the plasmids and that the transfection ratio should be greater than or equal to 2/1 (figure 2a).

The enzyme protection assay showed that the naked plasmid was completely degraded by DNase I. Moreover, at an N/P ≥ 2 and after DNase I addition, the plasmid was not degraded and still gathered in the hole. Figure 2b shows that SPCIONPs also protected the plasmids from degradation by DNase.

## 3.3. Prussian blue assay

Figure 3 shows Prussian blue staining after 4 h of incubation with SPCIONPs/pIGFBP$_5$. In the SPCIONPs/pIGFBP$_5$ group, most of the blue particles were concentrated on the cell membrane, and a small number of blue particles were visible in the cells. No significant difference in cell morphology was observed compared to the control group, indicating that SPCIONPs had no obvious cytotoxicity within 4 h.

## 3.4. CCK8 assay

A CCK8 assay was used to detect the toxicity of PUC19, CS/PUC19, SPCIONPs/PUC19, and PEI/PUC19. Figure 4 shows that the growth of the cells in the PUC19, CS/PUC19, and SPCIONPs/PUC19 groups at 24 h was not inhibited compared to the control group, and the cell survival rate in the SPCIONPs/PUC19 group was even greater than that in the control group, but the difference was not statistically significant. The cell survival rate in the PEI/PUC19 group was 86.59 ± 0.05%, which was significantly different from that in the SPCIONPs/PUC19 group.

After 48 h of the incubation with PUC19, CS/PUC19 or SPCIONPs/PUC19, the cell survival rate was not significantly different from that in the control group, whereas the cell survival rate in the PEI/PUC19 group was 73.37 ± 0.06%, which was also significantly different from that in the SPCIONPs/PUC19 group. No significant difference was observed between the CS/PUC19 and SPCIONPs/PUC19 groups at 24 or 48 h. The overall trend of the live–dead staining assay was generally consistent with that observed in the CCK8 assay. As shown in figure 5, red fluorescence represented dead cells, and green

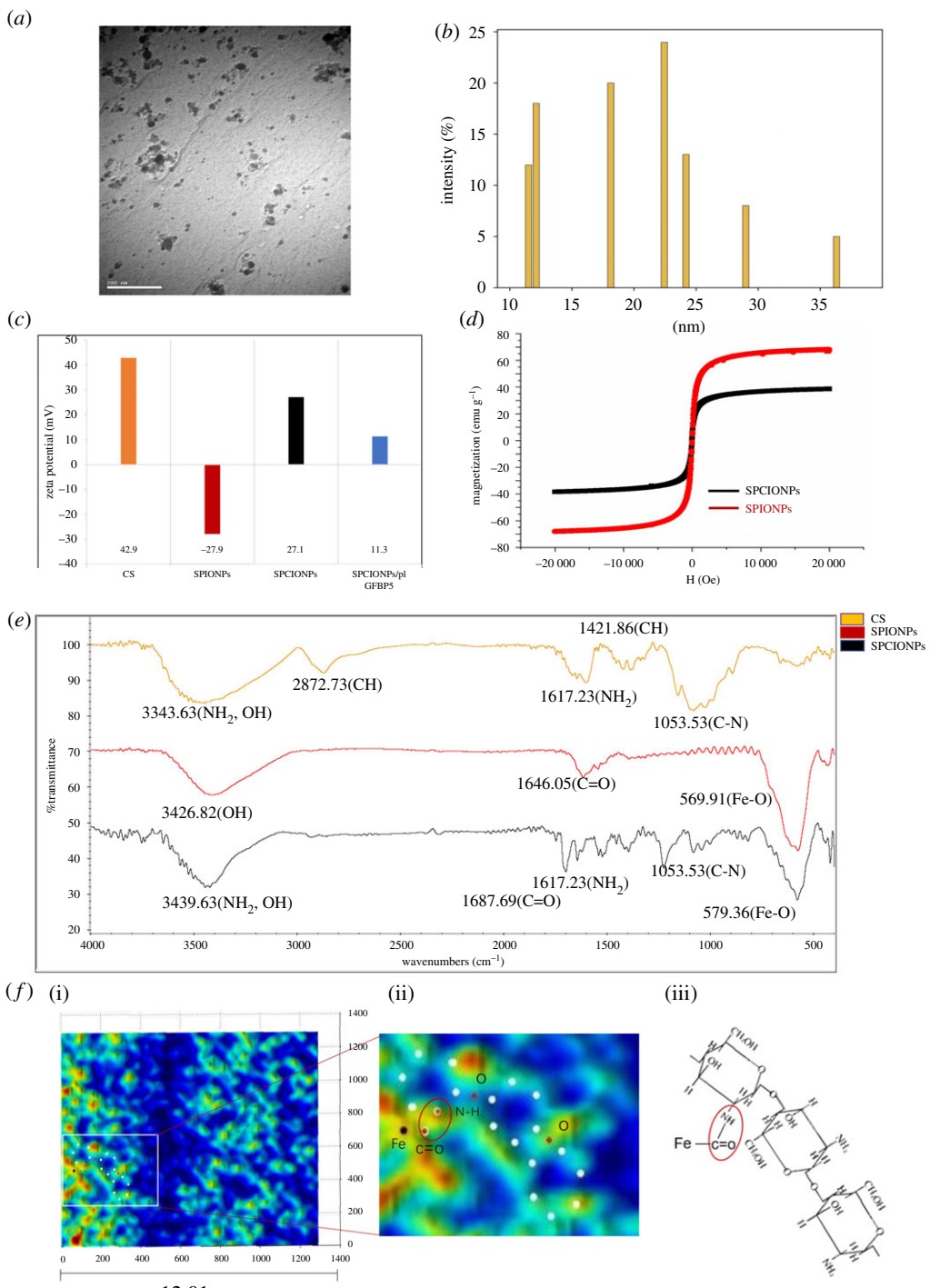

**Figure 1.** Characterization of SPIONPs and SPCIONPs. (*a*) TEM image of SPCIONPs; (*b*) the histogram to comment on size of SPCIONPs; (*c*) zeta potential of CS, SPIONPs, SPCIONPs and SPCIONPs/pIGFBP₅; (*d*) changes in magnetization curves of SPIONPs and SPCIONPs; (*e*) FT-IR spectra of CS, SPIONPs, and SPCIONPs; (*f*) (i) an AFM image; (ii) a partially enlarged image; (iii) molecular structure diagram.

fluorescence represented live cells. The number of red cells increased with time, and the number of red cells in the PEI/PUC19 group was the highest at the same time point.

## 3.5. Cell uptake assay and *in vitro* transfection

Figure 6 shows that FITC-CS/pIGFBP₅ and FITC-SPCIONPs/pIGFBP₅ were engulfed in endosomes, transported into the cells through endocytosis within 6 h after transfection and accumulated in large

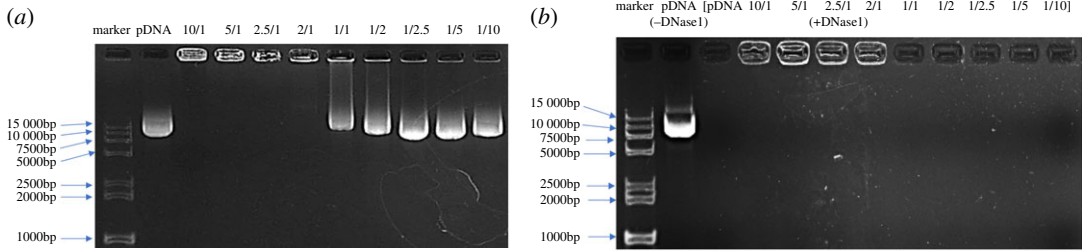

**Figure 2.** Agarose gel electrophoresis. (*a*) The combination of SPCIONPs and plasmids with different N/P ratios; (*b*) the combination of SPCIONPs and plasmids with different N/P ratios after DNase I addition.

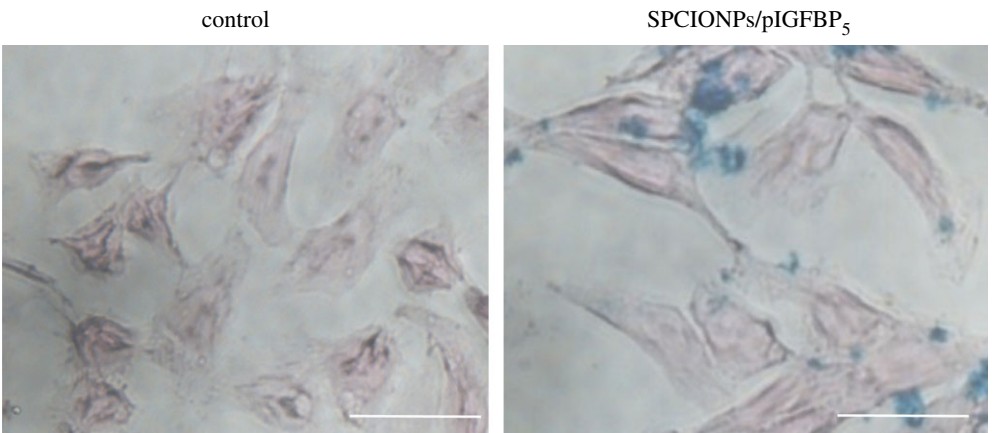

**Figure 3.** Prussian blue staining after transfection for 4 h. The scale bar is 100 μm.

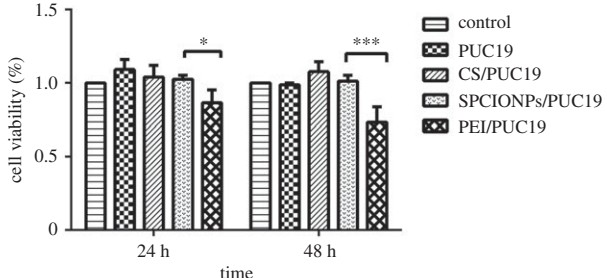

**Figure 4.** Cell survival was measured using a CCK8 assay. $^*p < 0.05$; $^{**}p < 0.01$; $^{***}p < 0.001$.

quantities around the nucleus. The FITC-SPCIONPs/pIGFBP$_5$ group showed more green fluorescent particles around the nucleus. At 24 h, both groups had red fluorescence expression of the plasmids, while the red fluorescence expression in the FITC-SPCIONPs/pIGFBP$_5$ group was greater than that in the FITC-CS/pIGFBP$_5$ group.

As shown in figure 7, the red fluorescence expression in the PEI/pIGFBP$_5$ group (positive control group) was the highest of all groups. The red fluorescence expression in the SPCIONPs/pIGFBP$_5$ group at 24 h after transfection was more than that in the CS/pIGFBP$_5$ group, which was more than that in the control group. This result was consistent with the laser confocal microscopy result.

# 4. Discussion

In this study, SPCIONPs were prepared to load the IGFBP$_5$ plasmids, which could inhibit lung metastasis of osteosarcoma. The gene carrier was composed of carboxylated SPIONPs and CS. The surface of carboxylated modified SPIONPs was rich in carboxyl groups and can be recombined with amino-rich chitosan in the form of covalent bond. In many studies, chlorosulfonic acid supported piperidine-4-carboxylic acid (PPCA) [34], citric acid [27,28], PEG [35], oleic acid [36] and polyamide acid [37] were used to modify SPIONPs to make its surface touch carboxyl groups, which can be combined with

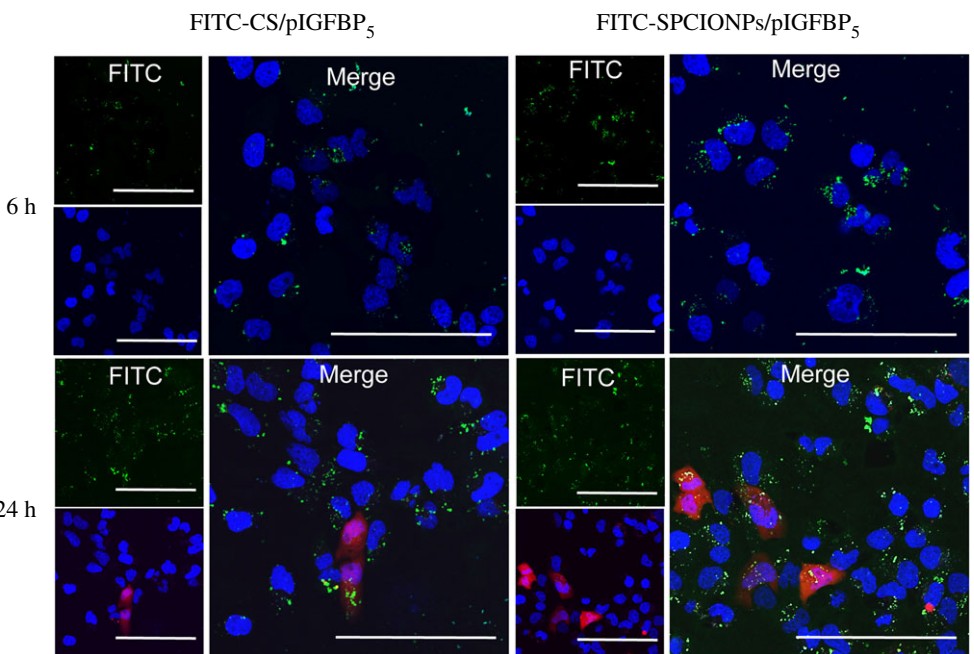

**Figure 5.** Cell survival was measured by a live–dead assay. (*a*) Red fluorescence represents dead cells, and green fluorescence represents live cells; (*b*) Quantitative analysis of cell viability. $^*p < 0.05$; $^{**}p < 0.01$; $^{***}p < 0.001$; the scale bar is 100 μm.

**Figure 6.** Nanoparticle complex uptake by cells after transfection for 6 and 24 h. Green fluorescence represents FITC-SPCIONPs/pIGFBP$_5$ and FITC-CS/pIGFBP$_5$, blue fluorescence indicates Hoechst-stained nucleus, and red fluorescence represents pIGFBP$_5$ expression. The scale bar is 100 μm.

different substances to show different functions. In our study, citric acid-modified SPIONPs were successfully prepared by chemical coprecipitation method. FTIR (figure 1*e*) showed that there were carboxyl absorption peaks and Fe–O absorption peaks on the surface of SPIONPs, indicating that we have successfully prepared citric acid modified superparamagnetic nanoparticles.

The carboxyl groups on the surface of SPIONPs and the amino groups of CS were connected by amide bonds through dehydration condensation. AFM and FTIR were used to detect the formation of

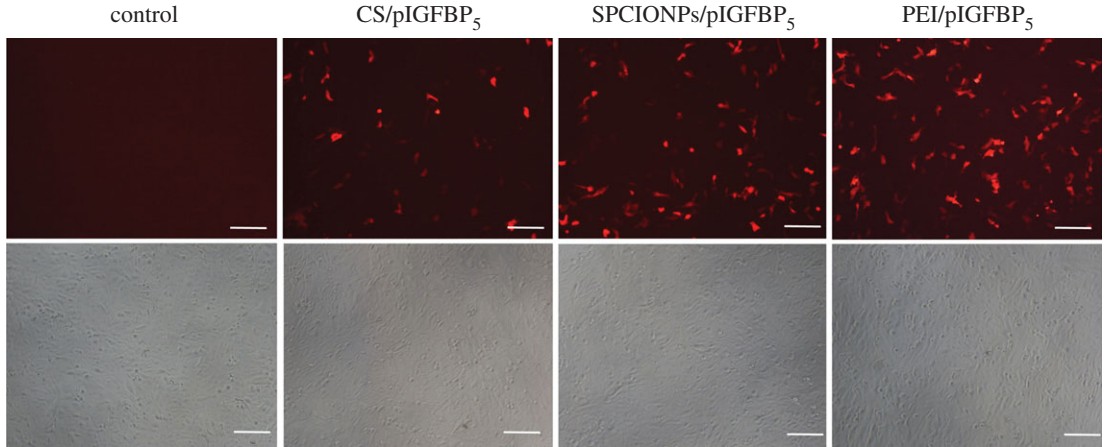

control | CS/pIGFBP$_5$ | SPCIONPs/pIGFBP$_5$ | PEI/pIGFBP$_5$

**Figure 7.** *In vitro* transfection of 143B cells. Transfection image under a fluorescence microscope at 100× magnification. The scale bar is 100 µm.

amide bond (figure 1$e,f$). CS is a polycation due to its amino groups, which are ionized in weakly acidic environments, allowing the polymer to interact with negatively charged surfaces, such as cell membranes [38,39]. After CS was successfully grafted onto SPIONPs, they could bind a large number of concentrated plasmids and effectively protect them from DNA enzyme degradation (figure 2), which was one of the necessary conditions for cell transfection [40]. Because the zeta potential of SPCIONPs/pIGFBP$_5$ was 11.3 mV, the positively charged SPCIONPs/pIGFBP$_5$ easily bound to the negatively charged cell membrane (figure 3), as shown by Prussian blue staining, which was the initial condition of cell transfection [41].

The toxicity of gene vectors is an important parameter of gene release systems [38]. Reduced toxicity is a significant advantage of CS-modified carriers, and CS has been shown to have low toxicity in many *in vitro* experiments [33,42,43]. Unsoy *et al*. [29] showed that CS-encapsulated superparamagnetic iron oxide nanoparticles (CS-MNPs) had no obvious cytotoxicity to cells, and when CS-MNPs were applied at high doses (1000 µg ml$^{-1}$) to HeLa cells, cell proliferation was reduced by only 2–5%. This observation is consistent with our research results. In this study, CCK8 and live–dead staining assays showed no significant difference between the SPCIONPs/PUC19 group and control group at 24 and 48 h (figures 4 and 5). Therefore, the SPCIONPs prepared in this study can serve as a non-viral vector with good biocompatibility and biosafety.

It is crucial to improve the transfection efficiency of CS as a gene vector system. CS is widely used in various fields due to its good biosafety. However, as a non-viral vector, chitosan does not have a strong proton-buffering effect, making it unable to escape from endosomes in a timely manner, leading to a lower transfection rate [40]. To increase the transfection rate of CS, Xu *et al*. [17] generated Arg-CS/pBMP-2 nanoparticles with arginine-linked CS to transfect preosteoblasts, Nam & Nah. [44] used CS-linked PEI to transfect colon cancer (HCT119) cells, and Wang *et al*. [45] prepared FA-PEG-CS to transfect liver cancer (HepG2) cells. However, magnetic transfection was used in this study, which forced the transfection vector and the target cell to stay in contact. In addition, the oscillation induced by magnetic transfection on the target cell surface can cause the nanoparticle complex to move, which helps increase the transfection rate by mechanically stimulating endocytosis [30,46,47]. Other researchers have shown that magnetic transfection can effectively increase the transfection rate of gene vectors [47,48]. Our *in vitro* transfection result showed that at 24 h, red fluorescence expression was greater in the SPCIONPs/pIGFBP$_5$ group than that in the CS/pIGFBP$_5$ group (figure 7). Laser confocal microscopy result showed (figure 6) that the FITC-SPCIONPs/pIGFBP$_5$ group exhibited greater red fluorescence expression with magnetic transfection because more nanoparticles accumulated around the nucleus. These findings confirm that magnetic transfection can promote transfection through targeted physical adsorption.

# 5. Conclusion

We successfully prepared carboxylated SPIONPs with a negative surface charge, which can combine with CS to form a non-viral vector called SPCIONPs. SPCIONPs have good biosafety, biocompatibility and can be tightly loaded with IGFBP$_5$ plasmids. Magnetic transfection can effectively promote

transfection of IGFBP$_5$ into 143B cells. Therefore, this magnetic transfection system has great potential in gene therapy for lung metastasis of osteosarcoma.

Data accessibility. Data available from the Dryad Digital Repository: https://doi.org/10.5061/dryad.7m0cfxprr [49].

Authors' contributions. Y.T. carried out the molecular laboratory work, participated in data analysis and drafted the manuscript; X.Q. and D.J. participated in the design of the study; J.W., Y.Z. and L.J. carried out the statistical analyses and critically revised the manuscript. All authors gave final approval for publication and agree to be held accountable for the work performed therein.

Competing interests. The authors declare no competing interests.

Funding. This work was supported by the medical research project of the Chongqing Science and Technology Commission (grant no. cstc2018jcyjAXO133) and Chongqing Municipal Health and Family Planning Commission (grant no. 2017MSXM048).

Acknowledgements. The authors thank Cong Luo from Children's Hospital of Chongqing Medical University for his great help in providing the laboratory for the preparation of superparamagnetic chitosan nanoparticles.

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
