## [Reviewer comments · Royal Society Open Science]

Review History

RSOS-201331.R0 (Original submission)

Review form: Reviewer 1

Is the manuscript scientifically sound in its present form?

Yes

Are the interpretations and conclusions justified by the results?

No

Is the language acceptable?

Yes

Do you have any ethical concerns with this paper?

No

Have you any concerns about statistical analyses in this paper?

No

Recommendation?

Accept with minor revision (please list in comments)

Comments to the Author(s)

See attached (Appendix A).

Review form: Reviewer 2**Is the manuscript scientifically sound in its present form?**

Yes

Are the interpretations and conclusions justified by the results?

Yes

Is the language acceptable?

No

Do you have any ethical concerns with this paper?

No

Have you any concerns about statistical analyses in this paper?

No

Recommendation?

Major revision is needed (please make suggestions in comments)

Comments to the Author(s)

Other comments

1. The author has not followed the RSOS format when writing the Abstract.
2. In this manuscript they have provided an AFM image to confirm the presence of chitosan on Fe surface. However, this needs a tedious procedure with the use of a high-resolution AFM. (Refer: <https://www.nature.com/articles/ncomms8766>). This study only provides little to no description of the procedure therefore difficult to assess. And also, not clear what is the distance units given in the AFM image (Å or nm). Even if the unit is Angstroms, the feature is still big to be a molecule as C-C bond is 1.5 Å.
2. What is abbreviated as "A "and "B" mixtures of live-dead staining in Page 6 line 60 and thereafter?
3. Typos in Page 7 line 5 (superscript is required when notating the units of wave number).
4. Figure 1, figure captions are too small
5. Article has also missed important citations (R. Soc. open sci. 5: 181369, European Journal of Pharmaceutics and Biopharmaceutics 117 (2017) 29–38, European Journal of Pharmaceutics and Biopharmaceutics, volume 128, July 2018, Pages 18-26, Chemistry Central Journal (2018) 12:119, WIREs Nanomed Nanobiotechnol. Volume12, Issue3, 2019:e1605)

Decision letter (RSOS-201331.R0)

Dear Professor Jiang:

Title: Magnetic transfection with superparamagnetic chitosan-loaded IGFBP5 nanoparticles and their in vitro biosafety
Manuscript ID: RSOS-201331

The editor assigned to your manuscript has now received comments from reviewers. We would like you to revise your paper in accordance with the referee and Subject Editor suggestions which can be found below (not including confidential reports to the Editor). Please note this decision does not guarantee eventual acceptance.

Please submit your revised paper before 15-Oct-2020. Please note that the revision deadline will expire at 00.00am on this date. If we do not hear from you within this time then it will be assumed that the paper has been withdrawn. In exceptional circumstances, extensions may be possible if agreed with the Editorial Office in advance. We do not allow multiple rounds of revision so we urge you to make every effort to fully address all of the comments at this stage. If deemed necessary by the Editors, your manuscript will be sent back to one or more of the original reviewers for assessment. If the original reviewers are not available we may invite new reviewers.

RSC Associate Editor:
Comments to the Author:
(There are no comments.)

RSC Subject Editor:
Comments to the Author:
(There are no comments.)

Reviewers' Comments to Author:
Reviewer: 1

Comments to the Author(s)
See attached

Reviewer: 2

Comments to the Author(s)
Other comments

1. The author has not followed the RSOS format when writing the Abstract.
2. In this manuscript they have provided an AFM image to confirm the presence of chitosan on Fe

surface. However, this needs a tedious procedure with the use of a high-resolution AFM. (Refer: <https://www.nature.com/articles/ncomms8766>). This study only provides little to no description of the procedure therefore difficult to assess. And also, not clear what is the distance units given in the AFM image (Å or nm). Even if the unit is Angstroms, the feature is still big to be a molecule as C-C bond is 1.5 Å.

2. What is abbreviated as "A" and "B" mixtures of live-dead staining in Page 6 line 60 and thereafter?
3. Typos in Page 7 line 5 (superscript is required when notating the units of wave number).
4. Figure 1, figure captions are too small
5. Article has also missed important citations (R. Soc. open sci. 5: 181369, European Journal of Pharmaceutics and Biopharmaceutics 117 (2017) 29–38, European Journal of Pharmaceutics and Biopharmaceutics, volume 128, July 2018, Pages 18-26, Chemistry Central Journal (2018) 12:119, WIREs Nanomed Nanobiotechnol. Volume12, Issue3, 2019:e1605)

Author's Response to Decision Letter for (RSOS-201331.R0)

See Appendix B.

RSOS-201331.R1 (Revision)

Review form: Reviewer 1

Is the manuscript scientifically sound in its present form?

Yes

Are the interpretations and conclusions justified by the results?

No

Is the language acceptable?

Yes

Do you have any ethical concerns with this paper?

No

Have you any concerns about statistical analyses in this paper?

No

Recommendation?

Accept with minor revision (please list in comments)

Comments to the Author(s)

After reading the revised manuscript, reviewers' comments, and the authors' responses, I think that reviewers' comments are not addressed properly, and this manuscript needs revision:

1. Abstract: Sentences are not complete or connected!
2. Introduction: After comment of Reviewer 2 (#2) regarding the novelty of this work, the authors responded that "We have improved the preparation method of CS-MNPs."! What is improved here?
First, if we compare particles in Figure 1a (this manuscript) and Figure 6B (Ref. 33), we can see individual and smaller particles in Ref. 33 compared to the aggregated particles in this study, so I do not see improvements in chitosan-coated iron oxide nanoparticles here. Second, the authors stated in Introduction "At present, there are many studies to modify Therefore, in this study, we prepared SPIONPs coated with CS to generate superparamagnetic chitosan-coated iron oxide"! It seems that this work is the first study to modify the surface of SPIONPs with CS, but it is not! Here, I expect the authors to refer to the publications on chitosan-coated iron oxide nanoparticles and state clearly what is the difference between this work and the previous works? and what is the novelty?
7. Experimental: If you used ferric chloride hexahydrate, you should change "ferrous chloride hexahydrate" to "ferric chloride hexahydrate" in section 2.1. Also, how many particles were counted to obtain the histogram (Figure 1b)? add this information to section 2.3.
8. Results: 1) X-axis & Y-axis in Figures 1b, c, d, e, and f(A) are not readable or clear. 2) There is a Chinese word in Figure 1c. 3) Figure 1a shows aggregated structure of SPCIONPs; what is the concentration of solution for TEM characterization; I suggest the authors to prepare dilute solution of SPCIONPs and re-do TEM characterization; they also need to provide TEM image of SPIONPs. 4) According to Figure 1a, SPCIONPs are aggregated; so how the authors counted 300 individual particles in such an aggregated system and provided histogram (Figure 1b)? 5) Provide DLS size distribution data for comparison with TEM data. 6) In FTIR data (Figure 1), peak around 3400 in SPIONPs corresponds to OH (not CH).

My main concern is size characterizations (TEM, DLS). Particle size reported here (21 nm) and previously (95 nm) are so different, which is a big deal in nanoscale (1-100 nm). Overall, I think the authors should do some size characterizations and rewrite some sections before resubmission.

Decision letter (RSOS-201331.R1)

Dear Professor Jiang:

Title: Magnetic transfection with superparamagnetic chitosan-loaded IGFBP5 nanoparticles and their in vitro biosafety
Manuscript ID: RSOS-201331.R1

Thank you for submitting the above manuscript to Royal Society Open Science. On behalf of the Editors and the Royal Society of Chemistry, I am pleased to inform you that your manuscript will be accepted for publication in Royal Society Open Science subject to minor revision in accordance with the referee suggestions. Please find the reviewers' comments at the end of this email.

The reviewers and handling editors have recommended publication, but also suggest some minor revisions to your manuscript. Therefore, I invite you to respond to the comments and revise your manuscript.

Because the schedule for publication is very tight, it is a condition of publication that you submit the revised version of your manuscript before 15-Nov-2020. Please note that the revision deadline will expire at 00.00am on this date. If you do not think you will be able to meet this date please let me know immediately.

- 1) A text file of the manuscript (tex, txt, rtf, docx or doc), references, tables (including captions) and figure captions. Do not upload a PDF as your "Main Document".
- 2) A separate electronic file of each figure (EPS or print-quality PDF preferred (either format should be produced directly from original creation package), or original software format)
- 3) Included a 100 word media summary of your paper when requested at submission. Please ensure you have entered correct contact details (email, institution and telephone) in your user account

4) Included the raw data to support the claims made in your paper. You can either include your data as electronic supplementary material or upload to a repository and include the relevant doi within your manuscript

5) All supplementary materials accompanying an accepted article will be treated as in their final form. Note that the Royal Society will neither edit nor typeset supplementary material and it will be hosted as provided. Please ensure that the supplementary material includes the paper details where possible (authors, article title, journal name).

Kind regards,
Dr Laura Smith
Publishing Editor, Journals

RSC Associate Editor:
Comments to the Author:
(There are no comments.)

RSC Subject Editor:
Comments to the Author:
(There are no comments.)

Reviewer comments to Author:
Reviewer: 1

Comments to the Author(s)
After reading the revised manuscript, reviewers' comments, and the authors' responses, I think that reviewers' comments are not addressed properly, and this manuscript needs revision:

1. Abstract: Sentences are not complete or connected!

2. Introduction: After comment of Reviewer 2 (#2) regarding the novelty of this work, the authors responded that “We have improved the preparation method of CS-MNPs.”! What is improved here?

First, if we compare particles in Figure 1a (this manuscript) and Figure 6B (Ref. 33), we can see individual and smaller particles in Ref. 33 compared to the aggregated particles in this study, so I do not see improvements in chitosan-coated iron oxide nanoparticles here. Second, the authors stated in Introduction “At present, there are many studies to modify Therefore, in this study, we prepared SPIONPs coated with CS to generate superparamagnetic chitosan-coated iron oxide”! It seems that this work is the first study to modify the surface of SPIONPs with CS, but it is not! Here, I expect the authors to refer to the publications on chitosan-coated iron oxide nanoparticles and state clearly what is the difference between this work and the previous works? and what is the novelty?

7. Experimental: If you used ferric chloride hexahydrate, you should change “ferrous chloride hexahydrate” to “ferric chloride hexahydrate” in section 2.1. Also, how many particles were counted to obtain the histogram (Figure 1b)? add this information to section 2.3.

8. Results: 1) X-axis & Y-axis in Figures 1b, c, d, e, and f(A) are not readable or clear. 2) There is a Chinese word in Figure 1c. 3) Figure 1a shows aggregated structure of SPCIONPs; what is the concentration of solution for TEM characterization; I suggest the authors to prepare dilute solution of SPCIONPs and re-do TEM characterization; they also need to provide TEM image of SPIONPs. 4) According to Figure 1a, SPCIONPs are aggregated; so how the authors counted 300 individual particles in such an aggregated system and provided histogram (Figure 1b)? 5) Provide DLS size distribution data for comparison with TEM data. 6) In FTIR data (Figure 1), peak around 3400 in SPIONPs corresponds to OH (not CH).

My main concern is size characterizations (TEM, DLS). Particle size reported here (21 nm) and previously (95 nm) are so different, which is a big deal in nanoscale (1-100 nm). Overall, I think the authors should do some size characterizations and rewrite some sections before resubmission.

Author's Response to Decision Letter for (RSOS-201331.R1)

See Appendix C.

Decision letter (RSOS-201331.R2)

Dear Professor Jiang:

Title: Magnetic transfection with superparamagnetic chitosan-loaded IGFBP5 nanoparticles and their in vitro biosafety
Manuscript ID: RSOS-201331.R2

It is a pleasure to accept your manuscript in its current form for publication in Royal Society Open Science. The chemistry content of Royal Society Open Science is published in collaboration with the Royal Society of Chemistry.

RSC Associate Editor
Comments to the Author:
(There are no comments.)

Reviewer(s)' Comments to Author:

Appendix A

In this work, the authors synthesized chitosan-coated iron oxide particles and characterized with FTIR, TEM, DLS, SQUID, and AFM. They also conducted experimental studies on their materials to investigate in vitro biosafety and use for cell transfection in the presence of magnetic transfection system. Here is my comments and questions:

- Title and Manuscript:** The authors used the word “*nanoparticles*”. Nanoparticles are routinely defined as particles with sizes in nanoscale (i.e., **between 1 and 100 nm**), exhibiting properties that are not found in their bulk counterparts. In the manuscript, the authors reported an average size of 95.60 nm in **Abstract** and approximately 100 nm in **Results - Characterization**. They also stated that “The average particle size of SPCIONPs detected by DLS and TEM was approximately 100 nm” (**Results – Characterization**)! Here are my comments on size characterization:
 - The TEM image (Fig. 1a) is not representative because it only shows one particle or I can say one aggregate of particles, so any conclusion on size based on only one particle is not acceptable! (*The best practices for reporting the size distributions of particles require transmission or scanning electron micrographs with histograms, in which at least 300 particles are measured per sample*), **so please provide a TEM image demonstrating more particles with a histogram to comment on size.**
 - What is the size range provided by DLS?
 - Finally, TEM and DLS do not provide the same information; TEM reports dry particle size as well as shape, and DLS provides solvated/hydrated diameter.
- Introduction:** Chitosan-coated iron oxide nanoparticles are not new and were synthesized even in the smaller size range in the past (Ref. 22 and 23)! **so please highlight what is novel about this work?**
- Experimental:** (1) Did you follow a literature procedure or modify any literature procedures to prepare the particles and perform biosafety studies? if yes cite those works in Experimental. (2) Did you use ferrous chloride hexahydrate or ferric chloride hexahydrate? Revise section 2.1 or 2.2 accordingly.
- Results - Characterization:** (1) A Scheme here could give visual understanding of your work. (2) Data on Fig. 1 don't have publication quality, and Fig. 1e is not readable. (3) I think the statement “AFM also verified that CS was bound to SPIONPs by amide bonds” is overinterpretation of AFM data; refer to other papers with such comments.
- Discussion:** Discussion is short, and there is not much discussion on materials characterization to make it interesting for chemistry readers.

This work is submitted under the subject category of Nanotechnology → Chemistry, but the way the manuscript is written and explained, I do not find this work interesting for chemistry or nanotechnology audience. **Therefore, I suggest the authors to revise some sections (introduction, results, and discussion) to make it more engaging for the chemistry audience.**

Appendix B

Dear Editor,

Thank you very much for your decision letter and reviewers' comments concerning our manuscript entitled "**Magnetic transfection with superparamagnetic chitosan-loaded IGFBP5 nanoparticles and their in vitro biosafety (RSOS-201331)**". Those comments are very valuable and helpful for revising and improving our manuscript, as well as the important guiding significance to our research. All of comments and suggestions for revision have been carefully taken care of in the revised manuscript. And our point-by-point responses to the reviewers' comments are as follows.

Revisions in the text are shown using **red highlight** for additions. In accordance with reviewer1's suggestions, we have revised the abstract and added the references. In accordance with reviewer2's suggestions, we have revised Figure1, introduction and discussion. We hope that the revisions in the manuscript and our responses will be sufficient to make our manuscript suitable for publication in Royal Society Open Science.

Thank you and best wishes.

Yours sincerely,

Yue Tang

Department of Traumatic Joint Center

The Third Affiliated Hospital of Chongqing Medical University(Gener Hospital)

Chongqing,401120 China

Phone: +8618323207854

E-mail: 386612312@qq.com

Reviewer1

1.The author has not followed the RSOS format when writing the Abstract.

Response: Thank you for your comments, which were very valuable and helpful for revising and improving our manuscript. According to your comments, we have re-written this part in the revised manuscript.

To study the application of superparamagnetic chitosan nanoparticles (SPCIONPs) as gene vectors using a magnetic transfection system for the targeted treatment of lung metastasis of osteosarcoma. The superparamagnetic chitosan nanoparticles were characterized by Transmission Electron Microscopy, Fourier Transform Infrared spectrometry, a Superconducting Quantum Interference Device and Atomic Force Microscopy. Their biosafety was determined by cell counting kit-8 (CCK8) and live-dead staining assays. In vitro transfection was detected by laser confocal microscopy. SPCIONPs, which can bind closely to plasmids and protect them from DNA enzyme degradation, were prepared with an average particle size of 21nm and zeta potential of 11.3 mV. The results of the CCK8 and live-dead staining assays showed that superparamagnetic chitosan nanoparticles loaded with Insulin-like growth factor-binding protein 5 (SPCIONPs/pIGFBP₅) induced no significant cytotoxicity compared to the control group. The in vitro transfection result suggested that pIGFBP₅ emitted a greater amount of red fluorescence in the SPCIONPs/pIGFBP₅ group than that in the chitosan-loaded IGFBP₅ (CS/pIGFBP₅) group. The prepared SPCIONPs had good biosafety and could be effectively used to transfer pIGFBP₅ into 143B cells, and they thus have good application prospects for the treatment of lung metastasis of osteosarcoma.

2.In this manuscript they have provided an AFM image to confirm the presence of chitosan on Fe surface. However, this needs a tedious procedure with the use of a high-resolution AFM. (Refer: <https://www.nature.com/articles/ncomms8766>). This study only provides little to no description of the procedure therefore difficult to assess. And also, not clear what is the distance units given in the AFM image (Å or nm). Even if the unit is Angstroms, the feature is still big to be a molecule as C-C bond is 1.5 Å.

Response: Thank you for your valuable advice.

1). The atomic force microscope was independently developed by Chongqing University. The working principle was to fix one end of a micro cantilever that was extremely sensitive to weak forces, and fixed a probe at the other end to sense the fluctuations of the sample surface through the repulsive force between atoms, and then measured the fluctuations of the cantilever to detect the surface of the sample through the STM probe. The molecular structure of prepared SPCIONPs was loaded onto the XY scanner of the AFM(IPC-208B, Chongqing University, China); the area to be scanned was localized on the monitor, and then images generated at room temperature under ambient conditions using the non-contact mode. A 100-mm scanner and a STM probe were used in the study, with a customized tungsten filament as the micro-cantilever to detect an area of 12.01 nm x 12.01 nm.

The atomic force microscopes made by Chongqing University were widely used in many studies.

References

Chen D, Gan H, Huang X, Shen Q, Du X, Tang W, Yang X. 2012 Effects of peripheral blood mononuclear cells morphology on vascular calcification in uremic patients on maintenance hemodialysis. *Ther Apher Dial.* **16.** 173-80.

(doi: 10.1111/j.1744-9987.2011.01044).

Li B, Wen M, Li W, He M, Yang X, Li S. 2011 Preparation and characterization of baicalin-poly-vinylpyrrolidone coprecipitate. *Int J Pharm.* **15.** 91-6.

(doi: 10.1016/j.ijpharm.2011.01.055).

Cen C, Wu J, Zhang Y, Luo C, Xie L, Zhang X, Yang X, Li M, Bi Y, Li T, He T. 2019 Improving Magnetofection of Magnetic Polyethylenimine Nanoparticles into MG-63 Osteoblasts Using a Novel Uniform Magnetic Field. *Nanoscale Res Lett.* **14.** 90.

(doi: 10.1186/s11671-019-2882-5)

2). In Figure 1, 1400 in the figure was a pixel not the scale, and the scale was converted to 12.01nm. So the unit was nm. The revised Figure was shown as follows.

3. What is abbreviated as “A” and “B” mixtures of live-dead staining in Page 6 line 60 and thereafter?

Response: Thank you for your comments. We used live-dead staining to detect the biosafety, there are three kinds of stains, which contained A (1mM Live-Dye), B (1mg/ml PI) and Staining buffer.

4. Typos in Page 7 line 5 (superscript is required when notating the units of wave number).

Response: Thank you for your valuable advice. We have revised the units of the wave number

The infrared spectra of CS, SPIONPs, and SPCIONPs in the wavenumber range of 400–4000 cm^{-1} were shown in Figure 1(e). The CS spectrum had unique amino (NH_2) characteristic peaks at 3343.63 cm^{-1} and 1617.23 cm^{-1} . The spectra of SPIONPs had characteristic COOH peaks, with C=O at 1646.05 cm^{-1} and O-H at 3426.82 cm^{-1} , the characteristic peaks of Fe_3O_4 and Fe-O were observed at 569.91 cm^{-1} . SPCIONPs exhibit the amide bond (NH-CO) formed by the amino group of chitosan and the

carboxyl group of SPIONPs, and the characteristic peaks of N-H (3439.63 cm^{-1} and 1617.23 cm^{-1}), C = O (1687.69 cm^{-1}) and Fe-O (579.36 cm^{-1}) were not affected. (Figure 1(e)).

4. Figure 1, figure captions are too small

Response: We apologize for the defect. We have revised the figure captions of Figure 1.

5. Article has also missed important citations (R. Soc. open sci. 5: 181369, European Journal of Pharmaceutics and Biopharmaceutics 117 (2017) 29–38, European Journal of Pharmaceutics and Biopharmaceutics, volume 128, July 2018, Pages 18-26,

Chemistry Central Journal (2018) 12:119, WIREs Nanomed Nanobiotechnol. Volume12, Issue3, 2019;e1605)

Response: Thank you for your references to supplement our article, which gave us good reference and provided great help for our article, and we have quoted them in the article. SPIONPs are widely used in biomedicine, including for targeted drug delivery [18,19,20,21], bioimaging, thermotherapy, photoablation therapy, biosensing, and thermal olfaction[22,23,24,25].

References

18. Manatunga DC, deSilva RM, deSilva KMN, deSilva N, Bhandari S, Yap YK, Costha NP. 2017 pH responsive controlled release of anticancer hydrophobic drugs from sodium alginate and hydroxyapatite bicoated iron oxide nanoparticles. *Eur J Pharm Biopharm.* **117**. 29-38. (doi: 10.1016/j.ejpb.2017.03.014)
19. Manatunga DC, de Silva RM, de Silva KMN, Malavige GN, Wijeratne DT, William GR. 2018 Effective delivery of hydrophobic drugs to breast and liver cancer cells using a hybrid inorganic nanocarrier: A detailed investigation using cytotoxicity assays, fluorescence imaging and flow cytometry. *Eur J Pharm Biopharm.* **128**. 18-26. (doi:10.1016/j.ejpb.2018.04.001)
20. Manatunga DC, de Silva RM, de Silva KMN, Wijeratne DT, Malavige GN, Williams G. 2018 Fabrication of 6-gingerol, doxorubicin and alginate hydroxyapatite into a bio-compatible formulation: enhanced anti-proliferative effect on breast and liver cancer cells. *Chem Cent J.* **12**:119. (doi: 10.1186/s13065-018-0482-6)
21. Manatunga DC, Godakanda VU, de Silva RM, de Silva KMN. 2019 Recent developments in the use of organic-inorganic nanohybrids for drug delivery. *Wiley Interdiscip Rev Nanomed Nanobiotechnol.* **12**. e1605. (doi: 10.1002/wnan.1605)

Reviewer2

1. Title and Manuscript: The authors used the word “nanoparticles”. Nanoparticles are routinely defined as particles with sizes in nanoscale (i.e., between 1 and 100 nm), exhibiting properties that are not found in their bulk counterparts. In the manuscript, the authors reported an average size of 95.60 nm in Abstract and approximately 100 nm in Results - Characterization. They also stated that “The average particle size of SPCIONPs detected by DLS and TEM was approximately 100 nm” (Results – Characterization)! Here are my comments on size characterization:

1) The TEM image (Fig. 1a) is not representative because it only shows one particle or I can say one aggregate of particles, so any conclusion on size based on only one particle is not acceptable! (The best practices for reporting the size distributions of particles require transmission or scanning electron micrographs with histograms, in which at least 300 particles are measured per sample), so please provide a TEM image demonstrating more particles with a histogram to comment on size.

Response: Thank you for your useful suggestions. We have provided a TEM image and a histogram to comment on size.

2) What is the size range provided by DLS?

Response: The particle size provided by DLS ranged from 37.8 nm to 295.3 nm.

3) Finally, TEM and DLS do not provide the same information; TEM reports dry particle size as well as shape, and DLS provides solvated/hydrated diameter.

Response: DLS mainly detected the average particle size of SPCIONPs in water, the solution in water lead to particle size agglomeration and lead to the increase of average particle size, while TEM mainly detected the morphology of SPCIONPs and the size of single particle.

2 Introduction: Chitosan-coated iron oxide nanoparticles are not new and were synthesized even in the smaller size range in the past (Ref. 22 and 23)! so please highlight what is novel about this work?

Response: Thank you for your comments. The average diameter of the single particle of the prepared SPCIONPs was about 21 nm, which was confirmed by TEM. We have improved the preparation method of CS-MNPs. Our team modified the superparamagnetic nanoparticles with citric acid to load carboxyl groups, and then recombined with amino-rich chitosan with amide bonds. The prepared SPCIONPs can

load plasmid to transfect cells.

3 Experimental: (1) Did you follow a literature procedure or modify any literature procedures to prepare the particles and perform biosafety studies? if yes cite those works in Experimental.

Response: Thank you for your valuable advice. Our team referred to and improved the method to prepare the carboxylated superparamagnetic nanoparticles[1,2], using ferric chloride hexahydrate, ferrous sulfate heptahydrate, and NaOH to prepare citric acid modified SPIONPs by coprecipitation method. We activated carboxyl group by EDC and NHS, then combined with chitosan by dehydration condensation method[3]. In this way, we have successfully prepared SPCIONPs, which was carboxylated superparamagnetic nanoparticles combined with chitosan through amide bond. We used CCK8 and live-dead staining assays to test the biosafety of the prepared SPCIONPs[4,5].

References

- [1] Antic B, Boskovic M, Nikodinovic-Runic J, Ming Y, Zhang H, Bozin ES, Janković D, Spasojevic V, Vranjes-Djuric S. 2017 Complementary approaches for the evaluation of biocompatibility of 90Y-labeled superparamagnetic citric acid (Fe,Er)₃O₄coated nanoparticles. *Mater Sci Eng C Mater Biol Appl.* 1. 157-164. (doi: 10.1016/j.msec.2017.02.023)
- [2] Patel U, Chauhan K, Gupte S. 2018 Synthesis, characterization and application of lipase-conjugated citric acid-coated magnetic nanoparticles for ester synthesis using waste frying oil. *3 Biotech* 8.211.(doi: 10.1007/s13205-018-1228-9)
- [3] Cen C, Wu J, Zhang Y, Luo C, Xie L, Zhang X, Yang X, Li M, Bi Y, Li T, He T. 2019 Improving Magnetofection of Magnetic Polyethylenimine Nanoparticles into MG-63 Osteoblasts Using a Novel Uniform Magnetic Field. *Nanoscale Res Lett.* 14. 90. (doi: 10.1186/s11671-019-2882-5)
- [4] Unsoy G, Yalcin S, Khodadust R, Gunduz G, Gunduz U. 2012 Synthesis optimization and characterization of chitosan-coated iron oxide nanoparticles produced for biomedical applications. *J Nanopart Res.* 14. 964. (doi: 10.1007/s11051-012-0964-8)
- [5] Barahuie F, Dorniani D, Saifullah B, Gothai S, Hussein MZ, Pandurangan AK, Arulselman P, Norhaizan ME. 2017 Sustained Release of Anticancer Agent Phytic Acid From Its Chitosan-Coated Magnetic Nanoparticles for Drug-Delivery System. *Int J Nanomedicine.* 12. 2361-2372.(doi: 10.2147/IJN.S126245)

(2) Did you use ferrous chloride hexahydrate or ferric chloride hexahydrate? Revise section 2.1 or 2.2 accordingly.

Response: Thank you for your valuable advice. We have revised section 2.2. We used ferric chloride hexahydrate and ferrous sulfate heptahydrate.

4.Results - Characterization: (1) A Scheme here could give visual understanding of your work. (2) Data on Fig. 1 don't have publication quality, and Fig. 1e is not readable.

Response: Thank you for your excellent and useful suggestion. SPCIONPs were successfully prepared, and their physicochemical characterizations are shown in Figure 1.

(3) I think the statement “AFM also verified that CS was bound to SPIONPs by amide bonds” is overinterpretation of AFM data; refer to other papers with such comments.

Response: Thank you for your suggestions. Our research team used similar methods to detect the amide bonds by AFM before, the reference was as follows.

Cen C, Wu J, Zhang Y, Luo C, Xie L, Zhang X, Yang X, Li M, Bi Y, Li T, He T. 2019 Improving Magnetofection of Magnetic Polyethylenimine Nanoparticles into MG-63 Osteoblasts Using a Novel Uniform Magnetic Field. *Nanoscale Res Lett.* 14. 90. (doi: 10.1186/s11671-019-2882-5)

5. Discussion: Discussion is short, and there is not much discussion on materials characterization to make it interesting for chemistry readers. This work is submitted under the subject category of Nanotechnology → Chemistry, but the way the manuscript is written and explained, I do not find this work interesting for chemistry or nanotechnology audience. Therefore, I suggest the authors to revise some sections (introduction, results, and discussion) to make it more engaging for the chemistry audience.

Response: Thank you for the comments. We have revised the introduction and discussion.

Introduction

Superparamagnetic iron oxide nanoparticles (SPIONPs) can act as a reagent with both active and passive targeting effects and can concentrate the carried drugs in the target tissue under the effect of an external magnetic field. Because of their advantages, such as biocompatibility, stability, environmental safety, and low price, SPIONPs are widely used in biomedicine, including for targeted drug delivery, bioimaging, thermotherapy, photoablation therapy, biosensing, and thermal olfaction. **At present, there are many studies to modify the surface of SPIONPs so that its surface is distributed in different groups, such as polymers, biomolecules, silica and metals which can provide the ensemble functional reactive group, e.g. aldehyde groups, hydroxyl groups, carboxyl groups and amino groups. Their groups can be linked to antibodies, proteins, DNA, enzymes and other bioactive substances for further application.** Therefore, in this study, we prepared carboxylated SPIONPs coated with CS to generate superparamagnetic chitosan-coated iron oxide nanoparticles (SPCIONPs) and conducted experimental studies to investigate their *in vitro* biosafety and use for cell transfection in the presence of magnetic transfection system.

Discussion

In this study, SPCIONPs were prepared to load the IGFBP₅ plasmids, which could inhibit lung metastasis of osteosarcoma. **The gene carrier was composed of carboxylated SPIONPs and CS. The surface of carboxylated modified SPIONPS was rich in carboxyl groups and can be recombined with amino-rich chitosan in the form of covalent bond. In many studies, citric acid, PEG, oleic acid, Chlorosulfonic Acid Supported Piperidine-4-carboxylic Acid (PPCA) and polyamide acid were used to modify SPIONPS to make its surface rich carboxyl groups, which can be combined with different substances to show different functions. In our study, Citric acid modified SPIONPS were successfully prepared by chemical coprecipitation method. FTIR (Figure 1(e)) showed that there were carboxyl absorption peaks and Fe-O absorption peaks on the surface of SPIONPS, indicating that we have successfully prepared citric acid modified superparamagnetic nanoparticles.**

The carboxyl groups on the surface of SPIONPs and the amino groups of CS were connected by amide bonds through dehydration condensation. AFM and FTIR were used to detect the formation of amide bond (Figure 1(e) and Figure 1(f)). CS is a polycation due to its amino groups, which are ionized in weakly acidic environments, allowing the polymer to interact with negatively charged surfaces, such as cell

membranes. After CS was successfully grafted onto SPIONPs, they could bind a large number of concentrated plasmids and effectively protect them from DNA enzyme degradation (Figure 2), which was one of the necessary conditions for cell transfection [26]. Because the zeta potential of SPCIONPs/pIGFBP₅ was 11.3 mV, the positively charged SPCIONPs/pIGFBP₅ easily bound to the negatively charged cell membrane (Figure 3), as shown by prussian blue staining, which was the initial condition of cell transfection.

Thank you again for your valuable comments to us, they are of important guiding significance for my future scientific research.

Appendix C

Dear Editor,

Thank you very much for your decision letter and reviewers' comments concerning our manuscript entitled "**Magnetic transfection with superparamagnetic chitosan-loaded IGFBP₅ nanoparticles and their in vitro biosafety (RSOS-201331.R1)**". Those comments are very valuable and helpful for revising and improving our manuscript, as well as the important guiding significance to our research. All of comments and suggestions for revision have been carefully taken care of in the revised manuscript. And our point-by-point responses to the reviewers' comments are as follows.

Revisions in the text are shown using **red highlight** for additions. We hope that the revisions in the manuscript and our responses will be sufficient to make our manuscript suitable for publication in Royal Society Open Science.

Thank you and best wishes.

Yours sincerely,

Yue Tang

Department of Traumatic Joint Center

The Third Affiliated Hospital of Chongqing Medical University(Gener Hospital)

Chongqing,401120 China

Phone: +8618323207854

E-mail: 386612312@qq.com

Reviewer1

1. Abstract: Sentences are not complete or connected!

Response: Thank you for your comments, which were very valuable and helpful for revising and improving our manuscript. According to your comments, we have re-written this part in the revised manuscript.

We prepared the superparamagnetic chitosan nanoparticles (SPCIONPs) to study the application of them as gene vectors using a magnetic transfection system for the targeted treatment of lung metastasis of osteosarcoma. The SPCIONPs were characterized by Transmission Electron Microscopy, Fourier Transform Infrared spectrometry, Superconducting Quantum Interference Device and Atomic Force Microscopy. Their biosafety was determined by cell counting kit-8 (CCK8) and live-dead staining assays. The transfection in vitro was detected by laser confocal microscopy. SPCIONPs, which can bind closely to plasmids and protect them from DNA enzyme degradation, were prepared with an average particle size of about 22nm and zeta potential of 11.3 mV. The results of the CCK8 and live-dead staining assays showed that superparamagnetic chitosan nanoparticles loaded with Insulin-like growth factor-binding protein 5 (SPCIONPs/pIGFBP₅) induced no significant cytotoxicity compared to the control group. The result of transfection in vitro suggested that pIGFBP₅ emitted a greater amount of red fluorescence in the SPCIONPs/pIGFBP₅ group than that in the chitosan-loaded IGFBP₅ (CS/pIGFBP₅) group. In conclusion, The prepared SPCIONPs had good biosafety and could be effectively used to transfer pIGFBP₅ into 143B cells, and they thus have good application prospects for the treatment of lung metastasis of osteosarcoma.

2. Introduction: After comment of Reviewer 2 (#2) regarding the novelty of this work, the authors responded that “We have improved the preparation method of CS-MNPs.”! What is improved here?

First, if we compare particles in Figure 1a (this manuscript) and Figure 6B (Ref. 33), we can see individual and smaller particles in Ref. 33 compared to the aggregated particles in this study, so I do not see improvements in chitosan-coated iron oxide nanoparticles here. Second, the authors stated in Introduction “At present, there are many studies to modify Therefore, in this study, we prepared SPIONPs coated with CS to generate superparamagnetic chitosan-coated iron oxide”! It seems that this work is the first study to modify the surface of SPIONPs with CS, but it is not! Here, I expect the authors to refer to the publications on chitosan-coated iron oxide nanoparticles and state clearly what is the difference between this work and the previous works? and what is the novelty?

Response: Thank you for your valuable advice.

The CS-MNPs in the ref.33 (Figure6B) were prepared by chemical co-precipitation and the CS and MNPs were combined with glycosidic bonds. However, the CS-MNPs we prepared by the same method in the reference literature were more agglomerated, so in our study, we divided the method of preparing SPCIONPs into two parts. First of all ,

SPIONPs modified with citric acid were prepared by co-precipitation method to increase the dispersibility and enrich the surface of SPIONPs with carboxyl groups, and then compound chitosan. The upper carboxyl groups of SPIONPs and chitosan were in the form of amide bonds. Also, our team has used the method of combining chitosan with carboxyl modified SPIONPS by amide bond in previous studies. The preparation of SPCIONPs was only a part of our research. Our main purpose was to use SPCIONPs to transfect IGFBP₅ into cells for further research.

7. Experimental: If you used ferric chloride hexahydrate, you should change “ferrous chloride hexahydrate” to “ferric chloride hexahydrate” in section 2.1. Also, how many particles were counted to obtain the histogram (Figure 1b)? add this information to section 2.3.

Response: Thank you for your comments.

We have revised “ferrous chloride hexahydrate” to “ferric chloride hexahydrate” in section 2.1

The size distribution was determined by measuring diameters of one hundred NPs randomly selected on the TEM micrographs and we have revised section 2.3

8. Results: 1) X-axis & Y-axis in Figures 1b, c, d, e, and f(A) are not readable or clear.

Response: Thank you for your comments, which were very valuable and helpful for revising and improving our manuscript.

The Figure uploaded in our contribution system was in jpg format, and the resolution will be reduced when converted to PDF format. I have modified the picture and improved the resolution of the picture

2) There is a Chinese word in Figure 1c.

Response: Thank you for your advice

We have revised the Figure 1c

3) Figure 1a shows aggregated structure of SPCIONPs; what is the concentration of solution for TEM characterization; I suggest the authors to prepare dilute solution of SPCIONPs and re-do TEM characterization; they also need to provide TEM image of SPIONPs..

Reponse: Thank you for your comments

We have prepared dilute solution of SPCIONPs and used ultrasonic dispersion and re-did TEM characterization.(Left:SPCIONPs Right:SPIONPs)

4) According to Figure 1a, SPCIONPs are aggregated; so how the authors counted 300 individual particles in such an aggregated system and provided histogram (Figure 1b)?

Reponse: Thank you for your comments

We re-did TEM characterization and re-measured diameters of one hundred NPs randomly selected on the TEM micrographs

5) Provide DLS size distribution data for comparison with TEM data.

Reponse: Thank you for your comments

SPCIONPs detected by DLS and TEM were the same sample. Because the SPCIONPs in the liquid will lead to the agglomeration between the particles, the average particle size of SPCIONPs detected by DLS was 95.6 nm, which was larger than that of single nanoparticles detected by TEM.

6) In FTIR data (Figure 1), peak around 3400 in SPIONPs corresponds to OH (not CH)

Reponse: Thank you for your comments

We have revised the Figure 1e

My main concern is size characterizations (TEM, DLS). Particle size reported here (21

nm) and previously (95 nm) are so different, which is a big deal in nanoscale (1-100 nm). Overall, I think the authors should do some size characterizations and rewrite some sections before resubmission.

Response: Thank you for your valuable advice.

We have revised size characterizations. Particle size reported here (21 nm) and previously (95 nm) are so different. Because the SPCIONPs in the liquid will lead to the agglomeration between the particles, the average particle size of SPCIONPs detected by DLS was 95.6 nm, which was larger than that of single nanoparticles detected by TEM.